# Low-Temperature Argon Plasma Regulates Skin Moisturizing and Melanogenesis-Regulating Markers through Yes-Associated Protein

**DOI:** 10.3390/ijms22041895

**Published:** 2021-02-14

**Authors:** Hae-Young Kim, Gaurav Agrahari, Min Jung Lee, Lee-Jung Tak, Won-Kook Ham, Tae-Yoon Kim

**Affiliations:** Department of Dermato-Immunology, College of Medicine, Catholic University of Korea, Seoul 06591, Korea; ho-jh1122@hanmail.net (H.-Y.K.); agra.gaurav06@gmail.com (G.A.); princessmin@hanmail.net (M.J.L.); flwjd6744@naver.com (L.-J.T.); ham2326@naver.com (W.-K.H.)

**Keywords:** low-temperature argon plasma, skin moisturizing, skin barrier, melanogenesis, melanoma, yes-associated protein

## Abstract

Extensive water loss and melanin hyperproduction can cause various skin disorders. Low-temperature argon plasma (LTAP) has shown the possibility of being used for the treatment of various skin diseases, such as atopic dermatitis and skin cancer. However, the role of LTAP in regulating skin moisturizing and melanogenesis has not been investigated. In this study, we aimed to determine the effect of LTAP on yes-associated protein (YAP), a major transcriptional coactivator in the Hippo signaling pathway that is involved in skin moisturizing and melanogenesis-regulating markers. In normal human epidermal keratinocytes (NHEKs), the human epidermal keratinocyte line HaCaT, and human dermal fibroblasts (HDFs), we found that LTAP exhibited increased expression levels of YAP protein. In addition, the expression levels of filaggrin (FLG), which is involved in natural moisturizing factors (NMFs), and hyaluronic acid synthase (HAS), transglutaminase (TGM), and involucrin (IVL), which regulate skin barrier and moisturizing, were also increased after exposure to LTAP. Furthermore, collagen type I alpha 1 and type III alpha 1 (COL1A1, COL3A1) were increased after LTAP exposure, but the expression level of matrix metalloproteinase-3 (MMP-3) was reduced. Moreover, LTAP was found to suppress alpha-melanocyte stimulating hormone (α-MSH)-induced melanogenesis in murine melanoma B16F10 cells and normal human melanocytes (NHEMs). LTAP regulates melanogenesis of the melanocytes through decreased YAP pathway activation in a melanocortin 1 receptor (MC1R)-dependent manner. Taken together, our data show that LTAP regulates skin moisturizing and melanogenesis through modulation of the YAP pathway, and the effect of LTAP on the expression level of YAP varies from cell to cell. Thus, LTAP might be developed as a treatment method to improve the skin barrier, moisture content, and wrinkle formation, and to reduce melanin generation.

## 1. Introduction

The skin protects the body from trauma, regulates body temperature, and balances moisture and electrolytes. Moisture in the skin circulates from the blood to the dermal and epidermal layers, reduces transepidermal water loss, and increases the thickness and density of the skin to prevent aging factors such as pigmentation and wrinkles [1]. Repeated exposure to ultraviolet (UV) rays creates elasticity and wrinkles in the skin and causes dehydration of cells [2]. Dehydration of skin cells decreases skin barrier functions [3]. Natural moisturizing factor (NMF) synthesis-related proteins such as transglutaminase (TGM), filaggrin (FLG), and involucrin (IVL) play a pivotal role in skin moisturizing and barrier function [4,5,6]. Similarly, hyaluronic acid (HA) is known to increase moisture content in the skin by regulating hyaluronic acid synthase (HAS)-encoding genes [7]. TGM catalyzes the formation of isopeptide bonds between proteins to prevent water loss, and the main substrates of TGM are epidermal barrier proteins that form protein–lipid matrices such as FLG and IVL. The expression of the genes encoding these proteins increases with the formation of the epidermal barrier [4,5,6].

On exposure to the external environment, cytokines, and hormones, melanocytes which are present in the epidermal layer, produce melanin-producing factors, thereby forming melanin [8]. Melanin plays an important role in protecting skin cells from UV rays. However, UV ray-induced excessive melanin production is known to cause spots, freckles, and skin cancer [9,10]. The yes-associated protein (YAP) plays an important role in cell growth, proliferation, and tumor growth regulation; it is a key component of the Hippo pathway identified in *Drosophila* studies. YAP plays an important role in apoptosis, cell-to-cell contact inhibition, and stem cell self-renewal. Although YAP displays beneficial roles in tissue recovery and wound healing, over-activity of YAP was found to cause cancer [11,12].

Plasma refers to the fourth physical state of matter (along with solids, liquids, and gases). When energy is applied to a solid substance, it passes through a liquid state to become a gaseous state. If more energy is applied to such a gaseous substance, electrons are released from an atom or molecule and become a plasma state in which electrons and cations independently exist [13]. Plasma has broad applications, including in nanotechnology, biotechnology, and the development of new materials. It is reported that the application of plasma in the field of biomedical applications plays a role in observing, controlling, and transforming very small objects such as cells, atoms, and molecules in the body [13,14]. Many early experiments have shown that applying low-temperature plasma can sterilize various pathogenic bacteria, and LTAP has demonstrated expression of anti-aging genes in skin cells [14], anti-inflammatory effects in atopic dermatitis [15], and induction of natural death of LTAP-dependent skin cancer cells [16]. However, no studies have been conducted on the detailed mechanism of LTAP in skin moisturizing and regulating melanogenesis.

We hypothesized that LTAP may play an important role in regulating skin moisturizing and regulating melanogenesis. We tested this hypothesis using normal human epidermal keratinocytes (NHEKs), human epidermal keratinocyte line HaCaT, and human dermal fibroblasts (HDFs) with respect to skin moisturizing; and a murine melanoma cell line (B16F10) and normal human epidermal melanocytes (NHEMs) with respect to the regulation of melanogenesis. In addition, by observing the expression levels of YAP on LTAP exposure, we aimed to determine the mechanism of YAP in regulating skin moisturizing and melanin production. Our results showed that LTAP-mediated regulation of skin moisturizing and melanogenesisis was dependent on YAP.

## 2. Results

### 2.1. LTAP Induces Skin Barrier and Moisturizing Factors

We first investigated the effect of LTAP exposure for 1, 3, or 5 min on the skin barrier and NMF synthesis-related gene expression in HaCaT cells and NHEKs. On LTAP exposure, the expression levels of skin barrier and moisturizing-related genes, including TGM, FLG, IVL, and HAS-1, 2 were found to increase at the protein and transcript levels in both HaCaT cells and NHEKs (Figure 1A,B). The expression levels of these genes were highest on LTAP exposure for 5 min.

### 2.2. Effects of LTAP on the Skin Proliferation and Aging-Related Gene

Next, we aimed to determine the effect of LTAP exposure on proliferation and aging factors in HDFs. Transforming growth factor-β (TGF-β) is a cytokine that promotes collagen synthesis and regulates the vascular endothelial growth factor (VEGF)-encoding gene, which is involved in cell proliferation and skin tissue regeneration [17,18]. Here, we found that LTAP exposure increases the expression levels of TGF-β and Collagen type I/III, alpha 1 (COL1A1, COL3A1), and decreases the matrix metallopeptidase-3 (MMP-3) expression at protein level (Figure 2A). Moreover, LTAP exposure also increases the expression levels of TGF-β, VEGF and COL1A1, COL3A1, and decreases the expression level of MMP-3 at transcript level (Figure 2B).

### 2.3. The Skin Moisturizing and Aging Factors Are Mediated by YAP

To understand the mechanism by which LTAP affects skin moisturizing and aging factors, HaCaT cells, NHEKs, and HDFs were exposed to LTAP for 1, 3, or 5 min, and then further incubated for 24 h. As shown in Figure 3A, LTAP increased the activation of YAP protein and significantly higher phosphorylation levels were observed in the group exposed for 5 min compared with exposure for 1 or 3 min. Verteporfin (VP), an inhibitor of YAP, is a sensitizer approved by the U.S. Food and Drug Administration for use in photodynamic therapy for age-related macular degeneration [19]. HaCaT cells and HDFs were treated with VP at concentrations of 5 and 10 μM and incubated for 24 h. The expression levels of both the phosphorylated and total form of YAP were found to be inhibited with VP treatment (Figure 3B). In addition, the mRNA expression level of the skin barrier and moisturizing-related genes and factors related to aging were found to be suppressed on VP treatment in HaCaT cells and HDFs, respectively (Figure 3C,D).

### 2.4. YAP Overexpression or Knockdown in HaCaT Cells Regulates Skin Barrier and Moisturizing Factors

To further evaluate the YAP-mediated regulation of the skin barrier and moisture regulation, we transfected HaCaT cells with YAP plasmid and small interfering RNA (siRNA) and determined the expression levels of the skin barrier and moisturizing-related genes. As shown in Figure 4A,B, overexpression of YAP in HaCaT cells significantly increased the expression levels of TGM, FLG, IVL and HAS-1, 2. Conversely, the depletion of YAP by siRNA in HaCaT cells significantly reduced the expression levels of TGM, FLG, IVL and HAS-1, 2 (Figure 4C,D). Collectively, these data indicate that the amount of YAP expression in HaCaT cells regulates skin barrier and moisturizing-related genes.

### 2.5. LTAP Inhibits Melanin Production in B16F10 Cells and NHEMs

To understand the effect of LTAP on melanogenesis, we exposed B16F10 cells and NHEMs to LTAP for 1, 3, or 5 min. Our results showed that the melanin content in B16F10 cells was significantly decreased on LTAP exposure (Figure 5A,B). The main factors that produce melanin are microphthalmia-associated transcription factor (MITF), tyrosinase (TYR), and tyrosinase-related proteins-1, 2 (TYRP-1, TYRP-2) [20]. Here, we found that LTAP exposure significantly decreased the expression levels of MITF, TYR and TYRP-1, but not TYRP-2, in both B16F10 cells and NHEMs (Figure 5C). Furthermore, the melanin-producing factors other than TYRP-2 were significantly reduced at the protein level in cells exposed for 5 min compared with cells exposed for 1 or 3 min in both B16F10 cells and NHEMs (Figure 5D).

### 2.6. LTAP Inhibits Receptors and Genes Involved in Melanin Production and Melanoma Proliferation

The proliferation and differentiation of melanoblasts is regulated by regulatory factors, such as endothelin (EDN) and Wingless-type protein (WNT) [21], and melanocortin 1 receptor (MC1R) is reported to be a major regulator of human pigmentation and a melanoma-related gene [22]. To elucidate the mechanism of the anti-melanogenesis effect of LTAP, we exposed B16F10 cells to LTAP and determined the expression levels of MC1R, EDN1, and WNT by Western blot analysis. As shown in Figure 6A, exposure to LTAP inhibited the expression of MC1R, EDN1, and WNT. Alpha-melanocyte stimulating hormone (α-MSH) is a ligand of MC1R that is known to activate the downstream protein kinase A (PKA), cyclic adenosine monophosphate (cAMP), and mitogen-activated protein kinase (MAPK) signaling pathways [23,24]. Here, we found that LTAP exposure suppressed the α-MSH-mediated activation of downstream PKA, cAMP response element-binding protein (CREB), and MAPK signaling, including of p38, extracellular signal-regulated kinase (ERK), and c-Jun-N-terminal kinase (JNK) (Figure 6B,C). In addition, to elucidate the anti-proliferative mechanism of LTAP, we determined the expression levels of cell cycle-related proteins. We found that LTAP exposure down-regulated the expression of Cyclin A, Cyclin B1, Cyclin D1, Cyclin-dependent kinase 1 (CDK1), and CDK2 (Figure 6D).

### 2.7. Melanogenesis Related Genes Are Regulated by Verteporine on YAP Inhibitors

Next, we conducted an experiment to find out the association between LTAP and YAP in B16F10 and NHEM cells. Interestingly, we found that exposure of LTAP inhibited the expression levels of YAP in both B16F10 and NHEM cells (Figure 7A). In addition, VP, an inhibitor of YAP, was found to downregulate the expression of YAP and melanogenesis regulating factors such as MC1R and MITF in B16F10 cells (Figure 7B). To further elucidate the inhibitory mechanism, we investigated the effect of VP on the *SRY (sex determining region Y)-box 10 (SOX10)* and *paired box gene 3 (PAX3)* transcription factors that regulate melanoma markers such as *protein melan-A (MART-1)* and *MITF* expressed in melanoma cells [25,26,27]. Our result showed that the gene expression levels of *MC1R, MITF, MART-1, SOX10* and *PAX3* were all significantly suppressed by VP in a concentration-dependent manner (Figure 7C). These results showed that the anti-melanogenesis effect of LTAP is mediated by YAP.

### 2.8. Overexpression and Knockdown of YAP Regulates Melanogenesis-Related Molecule Expression in B16F10 Cells

To confirm YAP-mediated regulation of melanogenesis-related molecules, we either over-expressed YAP in B16F10 cells from a plasmid or knocked down its expression using siRNA. As Figure 8A,B show, the overexpression of YAP in B16F10 cells significantly increased the expression of MITF, TYR and TYRP-1 at both proteins and mRNA levels. However, LTAP did not affect the expression level of TYRP-2. Knockdown of the *YAP* gene suppressed the mRNA and protein levels of MITF, TYR, and TYRP-1 but not TYRP-2 (Figure 8C,D). These data indicate that the amount of YAP expression in B16F10 cells and NHEMs regulates the growth of melanoma cells and melanogenic factors in melanocytes. Overall, the data suggests that LTAP may be an effective treatment for melanoma and hyperpigmentation through regulation of YAP.

## 3. Discussion

This study investigated the effects of LTAP exposure on the skin barrier, moisturizing function, and the mechanisms that potentially mediate this effect. As shown in Figure 1, exposure of HaCaT cells and NHEKs to LTAP increased the expression of skin barrier and moisturizing-related genes (TGM, FLG, IVL, and HAS-1, 2) that enhance moisturizing power. In addition, exposure of HDFs to LATP increased the expression of TGF-β, which is known to promote collagen synthesis, and VEGF, which is involved in cell proliferation and regeneration [18]. The expression level of MMP-3 was suppressed, and the expression levels of COL1A1 and COL3A1 were increased (Figure 2). These results suggest that LTAP up-regulates the expression of TGM, FLG, IVL, and HAS-1, 2 in keratinocytes, supplying moisture to the epidermal layer, which would help to maintain a stronger skin barrier and prevent skin aging and wrinkle formation.

It was reported that LTAP exposure improves the transdermal transduction of epidermal growth factors by regulating E-cadherin-mediated cell junctions [28]. In that study LTAP exposure inhibited the expression of E-cadherin, an adhesive molecule, in skin cells, inducing a temporary decrease in skin barrier function, thereby improving transdermal delivery of drugs and cosmetics. Inhibition of E-cadherin expression and reduction in skin barrier function were reported to be completely recovered within 3 h of LTAP exposure.

FLG is another protein involved in the skin barrier and has a function similar to that of E-cadherin, but the mechanism of interaction between the two molecules has not been accurately established. Indeed, much research has been done on E-cadherin and FLG, and there are conflicting reports about their interactions. In one study, a decrease in the expression of E-cadherins and occludin was reported to be due to FLG and loricrin deficiency; in another, inhibition of E-cadherin function induced FLG and loricrin expression [29,30]. For clear interpretation of our results, further experiments on the interaction of FLG and E-cadherin should be conducted.

The Hippo pathway plays an important role in controlling cell proliferation and size. YAP, an effector of the Hippo pathway, is expressed in the epidermis and is known to be an important gene for skin cell development and proliferation [31,32,33]. Studies show that YAP expression is significantly up-regulated during cutaneous wound healing and is localized to the keratinocyte nucleus around the wound [34], and that deletion of YAP reduced skin expansion [35,36]. It has been found that adhesion to the adhesion junction and the basement membrane plays an important role in the regulation of epidermal YAP activity [32,34]. VP, an inhibitor of YAP, has been reported to inhibit proliferation and induce G0/G1 arrest in pancreatic ductal adenocarcinoma cells, hepatocellular carcinoma cells, melanoma cells and leukemia cells [37,38,39,40]. However, few studies have focused on skin barrier and moisturizing-related and aging-related factors. Our data show that LTAP can enhance the activation of YAP protein, and the use of VP downregulated skin moisturizing and aging factors in skin cells (Figure 3). Similarly, overexpression of YAP stimulated the expression of skin barrier and moisturizing-related factors, whereas knockdown of YAP resulted in the downregulation of such genes (Figure 4). Taken together, our results indicate that skin barrier and moisturizing-related and aging-related genes are mediated by YAP, and that YAP plays an important role in maintaining skin homeostasis. Thus, we believe that LTAP can be considered as a therapy for cutaneous wound healing and maintenance of moisture content in skin through the regulation of YAP.

As mentioned above, YAP maintains skin homeostasis and regulates various types of cancer cells; the antitumor activity of VP has been reported in numerous studies [37,38,39,40]. Among them, malignant melanoma is the most fatal skin cancer, with a high risk of metastasis. There is evidence that YAP contributes to maintaining the transforming phenotype of melanoma cells [41,42], and VP contributes to the inhibition of melanoma cell proliferation [19,43]. However, little is known about the mechanism by which YAP regulates melanoma function. Here, we investigated the effects of LTAP exposure on YAP expression and anti-melanogenesis in B16F10 cells and NHEMs. LTAP exposure significantly inhibited mRNA and protein levels of the factors involved in anti-melanogenesis (MITF, TYR, and TYRP-1). However, it did not affect the expression level of TYRP-2 (Figure 5).

Melanogenesis in the skin is mediated through several melanogenesis signaling pathways, including MAPK signaling, and cAMP and protein kinase C-mediated pathways. Upregulation of intracellular cAMP activates PKA, which phosphorylates CREB, resulting in the increased expression of MITF [44]. In this study, we demonstrated that LTAP inhibited melanogenesis, its effect was mediated via the PKA, MAPK and CREB signaling pathways, and this effect inhibited cell proliferation (Figure 6). These results are in good agreement with the previously reported effects of microplasma on melanoma cell death, although the types of cells and plasma are different [16,45,46]. MITF is known as a central regulator of melanoma cell survival, proliferation, and differentiation [47,48], and the factors that regulate it include *MC1R, MART-1, SOX10* and *PAX3* [25,26,27]. Our data confirmed that LTAP exposure downregulated MITF and YAP at the protein level in both B16F10 cells and NHEMs. In addition, the expression levels of *MC1R*, *MART-1*, *SOX10* and *PAX3*, which are transcription factors that regulate MITF, were significantly suppressed by VP treatment in a concentration-dependent manner (Figure 7). In fact, it has been reported that YAP depletion reduces the expression of MITF [48], and we have specifically demonstrated a mechanism for the antitumor activity of VP. In addition, YAP overexpression and knockdown were investigated to determine the role of YAP in the factors related to melanogenesis in B16F10 cells. YAP overexpression resulted in a significant increase in the expression level of melanogenesis-related factors, whereas YAP knockdown resulted in significant down-regulation of melanogenesis-related factors at both the protein and mRNA levels (Figure 8). Taken together, our results demonstrate that LTAP mediates YAP to inhibit the proliferation of melanoma cells and regulate anti-melanogenesis activity. However, this was an in vitro system, and further research will be needed to see if these observations extend in vivo.

Our study investigated the role of an LTAP device in regulating skin moisture content, aging factors, melanoma cell proliferation and anti-melanogenesis in different types of skin cells. In summary, our results show that, as summarized in Figure 9, LTAP significantly increased the barrier, moisture content-related, and aging factors of skin cells, and suppressed melanogenesis in melanoma cell lines through YAP regulation. These results suggest that LTAP has the potential to be developed as an effective device to control skin moisturization, and for anti-melanogenesis.

## 4. Materials and Methods

### 4.1. LTAP Device

A microwave plasma generator was used for plasma exposure of cells and was kindly provided by Professor Gyu-Chun Kim of Pusan National University, Republic of Korea. For plasma generation, a 900 MHz coaxial transmission line resonator and ~2.5 standard L/min of argon gas were used together. The microwave plasma jet had a radius of 2.5 mm and a length of 7 mm, and the bulk power density of the plasma jet was estimated to be 14.55 W.mm^−3^. More information on this device can be found in ref [28].

### 4.2. Cell Culture and LTAP Exposure

Human HaCaT cells, HDFs, NHEKs, NHEMs and murine B16F10 cells were purchased from Thermo Fisher Scientific (Waltham, MA, USA). HaCaT cells, HDFs and B16F10 cells were cultured in Dulbecco’s modified Eagle’s medium (HyClone, GE Healthcare Life Sciences, Logan, UT, USA) with 10% fetal bovine serum (HyClone) and 1% penicillin–streptomycin (LS202-02, Welgene, Daegu, Korea). NHEKs and NHEMs were cultured in keratinocyte media (MEP1500CA, Gibco, Life Technologies, Carlsbad, CA, USA) and Medium 254 (M254500, Gibco) supplemented with human keratinocyte and melanocyte growth supplement (S0015, S0025, Gibco) and 1% penicillin-streptomycin at 37 °C in a humidified mixture of air (95% (*v*/*v*)) and 5% CO_2_. For LTAP exposure, 5 × 10^5^ cells were seeded in 35 mm dishes and incubated for 24 h. Before LTAP exposure, the medium was exchanged for 2 mL of new growth medium. The lid was opened, and the dish was placed 1 cm below the part where argon gas was ejected from the LTAP generator, and exposed for 1, 3, or 5 min. After exposure for the indicated time, the dishes were placed in a CO_2_ incubator at 37 °C.

### 4.3. VP Treatment, siRNA, and Plasmid DNA Transfection

The YAP inhibitor VP was purchased from Sigma Aldrich (SML0534, St. Louis, MO, USA), and siRNA targeting the *YAP* gene was purchased from Santa Cruz Biotechnology (sc-37007, 38637, 38638, Dallas, TX, USA). The *YAP* plasmid DNA (pcDNA4/HisMaxB-YAP) used for overexpression of *YAP* in cells was purchased from Addgene (Cambridge, UK). Cells were seeded into 6-well plates at a density of 2 × 10^5^ cells per well and at 70–80% confluency. They were transfected with 20 nM siRNA or 4 μg of *YAP* plasmid DNA using Lipofectamine 2000 transfection reagent (11668027, Invitrogen, Carlsbad, CA, USA) following the manufacturer’s instructions. After transfection for 24 h, cells were harvested for analysis.

### 4.4. Measurement of Melanin Content

Melanin content was analyzed as previously described [20]. Briefly, B16F10 cells and NHEMs were seeded at 2 × 10^5^ cells per well in 6-well plates for 24 h followed by exposure to LTAP for 1, 3, or 5 min. After 72 h, cells were homogenized with 1M NaOH solution (221465, Sigma Aldrich, St. Louis, MO, USA). The total melanin level was measured at an absorbance of 405 nm using a Spectra Max Plus spectrophotometer (BioTek, Winooski, VT, USA).

### 4.5. cDNA synThesis and Quantitative Real-Time PCR (qRT-PCR)

Total RNA was extracted from harvested cells using TRIzol reagent (15596018, Invitrogen), and cDNA was reverse transcribed from 1 μg total RNA using a QuantiTech Reverse Transcription Kit (205311, Qiagen, Hilden, Germany). PCR was performed using the QuantiTech SYBR Green PCR kit (204143, Qiagen) and Rotor-Gene 6000 (Corbett, Mortlake, NSW, Australia). The amplification program consisted of one cycle of 95 °C for 10 min, and 35 cycles of 95 °C for 20 s, 55 °C for 20 s, and 72 °C for 20 s. GAPDH (Glyceraldehyde 3-phosphate dehydrogenase) was used as an endogenous control for normalization. The changes in fold were determined by double delta CT analysis. Primers for PCR were synthesized by Bioneer (Daejeon, Korea). The primers used were as follows: human GAPDH, 5′-TGC ACC ACC AAC TGC TTA GC, 3′-GGC ATG GAC TGT CCT CAT CA; human YAP, 5′-TAG CCC TGC GTA GCC AGT TA, 3′-TCA TGC TTA GTC CAC TGT CTG T; human FLG, 5′-AGG GAA GAT CCA AGA GCC CA, 3′-ACT CTG GAT CCC CTA CGC TT; human IVL, 5′-GGT CCA AGA CAT TCA ACC AGC C, 3′-TCT GGA CAC TGC GGG TGG TTA T; human TGM, 5′-GAA ATG CGG CAG ATG ACG AC, 3′-AAC TCC CCA GCG TCT GAT TG; human HAS-1, 5′-CCA CCC AGT ACA GCG TCA AC, 3′-CAT GGT GCT TCT GTC GCT CT; human HAS-2, 5′-TTC TTT ATG TGA CTC ATC TGT CTC ACC GG, 3′-ATT GTT GGC TAC CAG TTT ATC CAA ACG; human TGF-β, 5′-TAC CTG AAC CCG TGT TGC TCT C, 3′-GTT GCT GAG GTA TCG CCA GGA A; human VEGF, 5′-TTG CCT TGC TGC TCT ACC TCC A, 3′-GAT GGC AGT AGC TGC GCT GAT A; human MMP-3, 5′-CAC TCA CAG ACC TGA CTC GGT T, 3′-AAG CAG GAT CAC AGT TGG CTG G; human COLA1A, 5′-GAT TCC CTG GAC CTA AAG GTG C, 3′-AGC CTC TCC ATC TTT GCC AGC A; human COLA3A1, 5′-TGG TCT GCA AGG AAT GCC TGG A, 3′-TCT TTC CCT GGG ACA CCA TCA G; human MITF, 5′-ACC TTC TCT TTG CCA GTC CA, 3′-TTG GGC TTG CTG TAT GTG GT; human TYR, 5′-CTT GTG AGC TTG CTG TGT CG, 3′-GTG AGG TCA GGC TTT TTG GC; human TYRP-1, 5′-ATG TCG CTC AGT GCT TGG AA, 3′-GAC TTC GAA CAG CAG GGT CA; human TYRP-2, 5′-AAC TCC CTT CCC TGC ATG TG, 3′-TTG TGA CCA TAG GGG CCA G; mouse GAPDH, 5′-CCA TGA CAA CTT TGG CAT TG, 3′-CCT GCT TCA CCA CCT TCT TG; mouse MITF, 5′-CTG ATC TGG TGA ATC GGA TC, 3′-TCC TGA AGA AGA GAG GGA GC; mouse TYR, 5′-GGG CCC AAA TTG TAC AGA GA, 3′-ATG GGT GTT GAC CCA TTG TT; mouse TYRP-1, 5′-AAG TTC AAT GGC CAG GTC AG, 3′-TCA GTG AGG AGA GGC TGG TT; mouse TYRP-2, 5′-CCC GAG GCA ACC AAC ATC T, 3′-AAG TTT CCT GTG CAT TTG CAT GTC; mouse MC1R, 5′-CTC CAC AGA CCG CTT CCT AC, 3′-ACA TAC AGG CAC CAA GGC TC; mouse MART-1, 5′-ACA GCC GAC TGT CTT CTC AAG, 3′-AGC ATT CTA AAG CGA AAC ACC G; mouse SOX10, 5′-CCC CTT CAT TGA GGA GGC TG, 3′-CCA GGT GGG CAC TCT TGT AG; mouse PAX3, 5′-AGC AAA CCC AAG CAG GTG AC, 3′-CGA TCA CAG ACA GCG TCC TT.

### 4.6. Western Blotting

Western blotting was performed as previously described [20]. Equal amounts of protein were separated by 10% sodium dodecyl sulfate-polyacrylamide gel electrophoresis (Elpis-biotech, Daejeon, Republic of Korea), and proteins were detected by enhanced chemiluminescence (Advansta, San Jose, CA, USA) using the LAS 3000 detection system (Fuji Film, Tokyo, Japan). Primary antibodies used were FLG (1:500; LifeSpan BioSciences, Seattle, WA, USA); IVL (1:500; BioLegend, San Diego, CA, USA); pYAP, CERB, p38, EKR, JNK, Cyclin B1, D1, E1, CKD2 (1:1000; Cell Signaling, Beverly, MA, USA); MMP-3, COL1A1, YAP, TYRP-1,2, PKA, Cyclin A, β-actin (1:500; Santa Cruz Biotechnology, Dallas, TX, USA); MC1R (1:500; Bioss Antibodies, Woburn, MA, USA); MITF, TYR, EDN1, WNT, and CDK1 (1:1000; Abcam, Cambridge, UK).

### 4.7. Statistical Analysis

The results of this study were repeated three or more times. Statistical significance was evaluated by one way analysis of variance, and the data represent the mean ± SD. Statistically significant differences were considered as *p* < 0.05 (* *p* < 0.05, ** *p* < 0.01, *** *p* < 0.001).

## Figures and Tables

**Figure 1 ijms-22-01895-f001:**
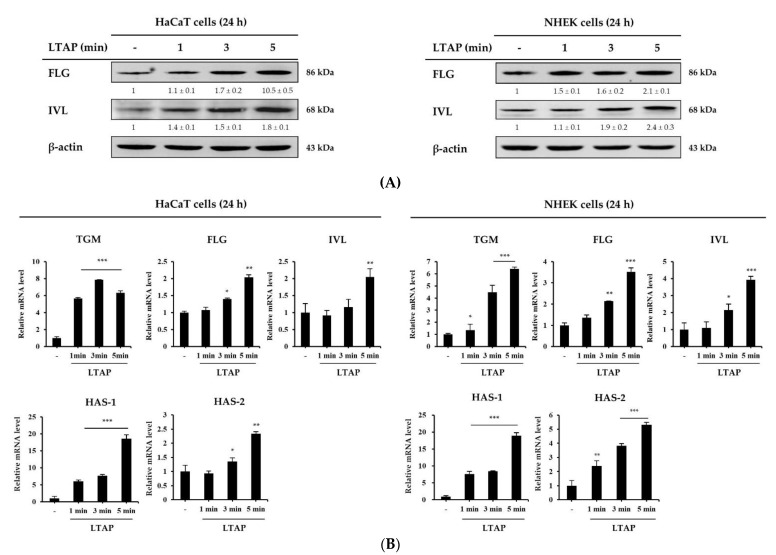
Effects of low-temperature argon plasma (LTAP) on the skin barrier and moisturizing factors in human epidermal keratinocyte cell line HaCaT and normal human epidermal keratinocytes (NHEKs). HaCaT cells and NHEKs were exposed to LTAP for 1, 3, or 5 min, and further incubated for 24 h. (**A**) Expression of filaggrin (FLG) and involucrin (IVL) was determined by Western blotting. Numerical values on the blots represent DU. The control was normalized to 1 DU. All data represent three independent experiments. (**B**) Expression of skin barrier and moisturizing-related genes (*tansglutaminases (TGM), FLG, IVL* and hyaluronic acid synthase-1, 2 *(HAS-1, 2*)) was measured by quantitative real-time PCR (qRT-PCR) in HaCaT cells and NHEKs. Results are expressed as the mean ± SD of three independent experiments. * *p* < 0.05, ** *p* < 0.01, *** *p* < 0.001 (control vs. LTAP-exposed cells).

**Figure 2 ijms-22-01895-f002:**
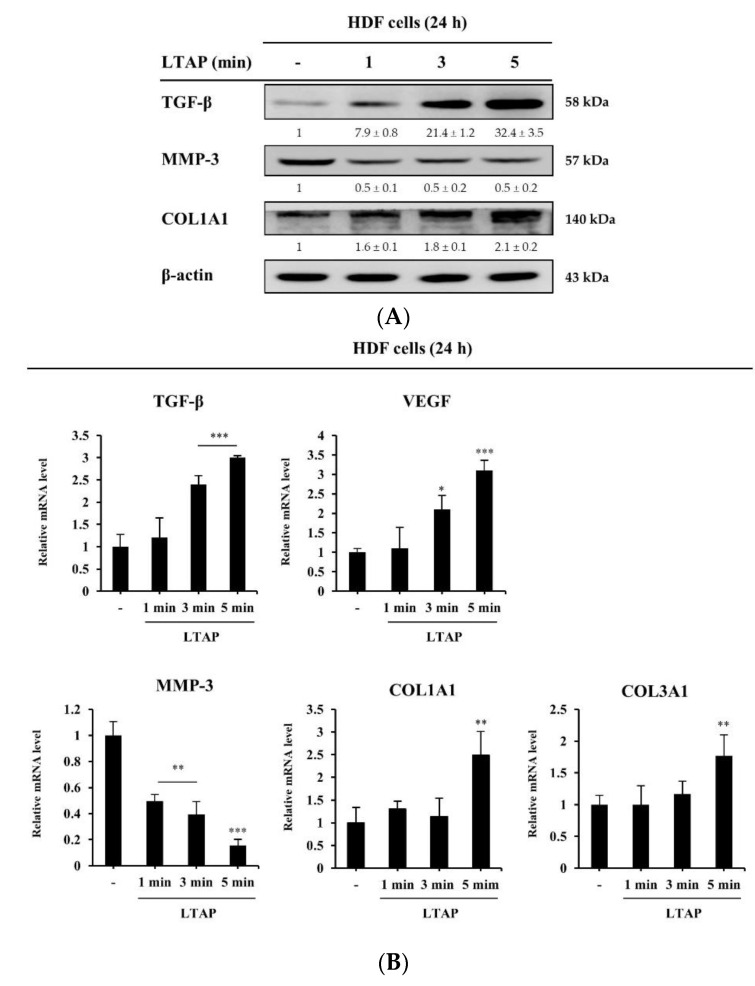
LTAP regulates proliferating and aging factors in human dermal fibroblasts (HDFs). HDFs were exposed to LTAP for 1, 3, or 5 min, and further incubated for 24 h. (**A**) Expression levels of transforming growth factor-β (TGF-β), matrix metallopeptidase-3 (MMP-3), and collagen type I alpha 1 (COL1A1) were analyzed by Western blotting. β-actin was used as a loading control. Numerical values on the blots represent DU. The control was normalized to 1 DU. All data represent three independent experiments. (**B**) mRNA expression analysis for *TGF-β*, *vascular endothelial growth factor* (*VEGF*), *MMP-3*, *COL1A1*, and *COL3A1* was performed by qRT-PCR. Data represent three independent experiments and are the means ± SD. * *p* < 0.05, ** *p* < 0.01, *** *p* < 0.001 (control vs. LTAP-exposed cells).

**Figure 3 ijms-22-01895-f003:**
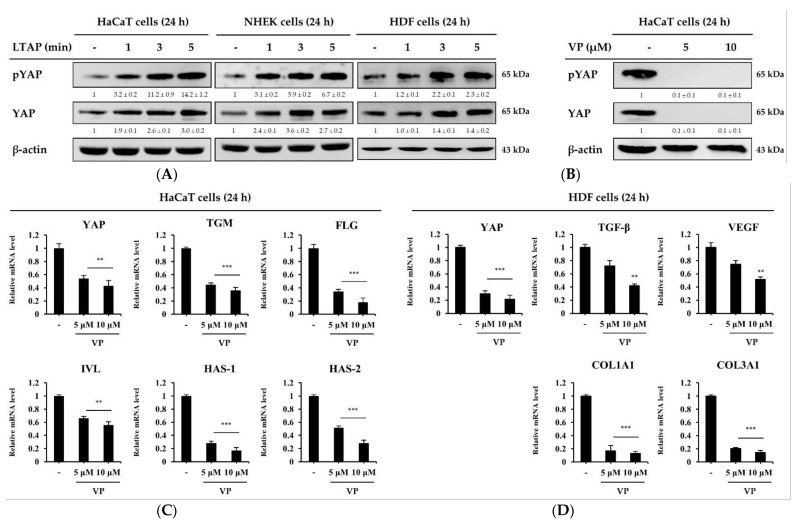
LTAP regulates skin moisturizing and aging factors through yes-associated protein (YAP). (**A**) HaCaT cells, NHEKs, and HDFs were exposed to LTAP for 1, 3, or 5 min, and further incubated for 24 h. Cells were harvested and the levels of phosphorylated YAP (pYAP), and YAP were evaluated by Western blotting. β-actin was used as a loading control. (**B**) HaCaT cells were treated with either 5 or 10 μM verteporin (VP), and further incubated for 24 h. The levels of pYAP and YAP were evaluated by Western blotting. Numerical values on the blots represent DU. The control was normalized to 1 DU. All data represent three independent experiments. (**C**) The expression of *YAP*, *TGM*, *FLG*, *IVL* and *HAS-1*, *2* in HaCaT cells was analyzed by qRT-PCR. (**D**) HDFs were treated with either 5 or 10 μM VP, and further incubated for 24 h. Cells were harvested for mRNA isolation, and expression of *YAP*, *TGF-β*, *VEGF*, *COL1A1* and *COL3A1* was determined by qRT-PCR. Data represent the mean ± SD. ** *p* < 0.01, *** *p* < 0.001. All data represent three independent experiments.

**Figure 4 ijms-22-01895-f004:**
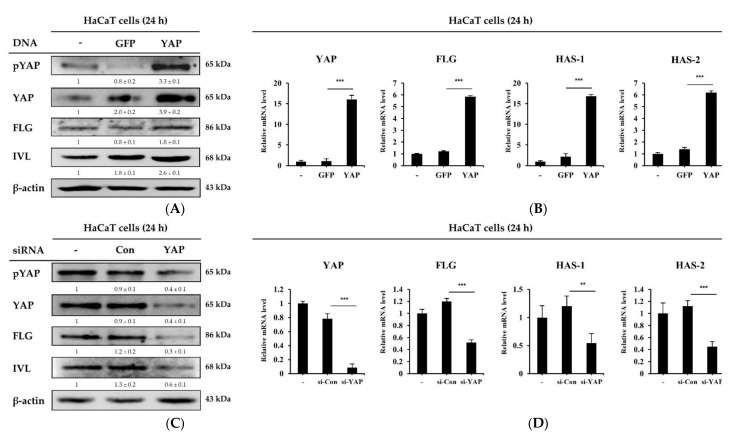
Skin barrier and moisturizing factors are mediated by YAP. (**A**,**B**) HaCaT cells were transfected with a plasmid overexpressing either green fluorescent protein (GFP; a control) or YAP, and incubated for 24 h. Cells were harvested and the expression of pYAP, YAP, FLG, and IVL was evaluated by Western blotting and the expression levels of *YAP*, *FLG*, and *HAS-1*, *2* were determined by qRT-PCR. For Western blotting, β-actin was used as a loading control. (**C**,**D**) HaCaT cells were transfected with control small interfering RNA (siRNA) or YAP siRNA, and further incubated for 24 h. The expression of pYAP, YAP, FLG, and IVL was determined at the protein level, and the expression of *YAP*, *FLG* and *HAS-1, 2* was analyzed at the mRNA level. Numerical values on the blots represent DU. The control was normalized to 1 DU. Data are the means ± SD and represent three independent experiments. ** *p* < 0.01, *** *p* < 0.001.

**Figure 5 ijms-22-01895-f005:**
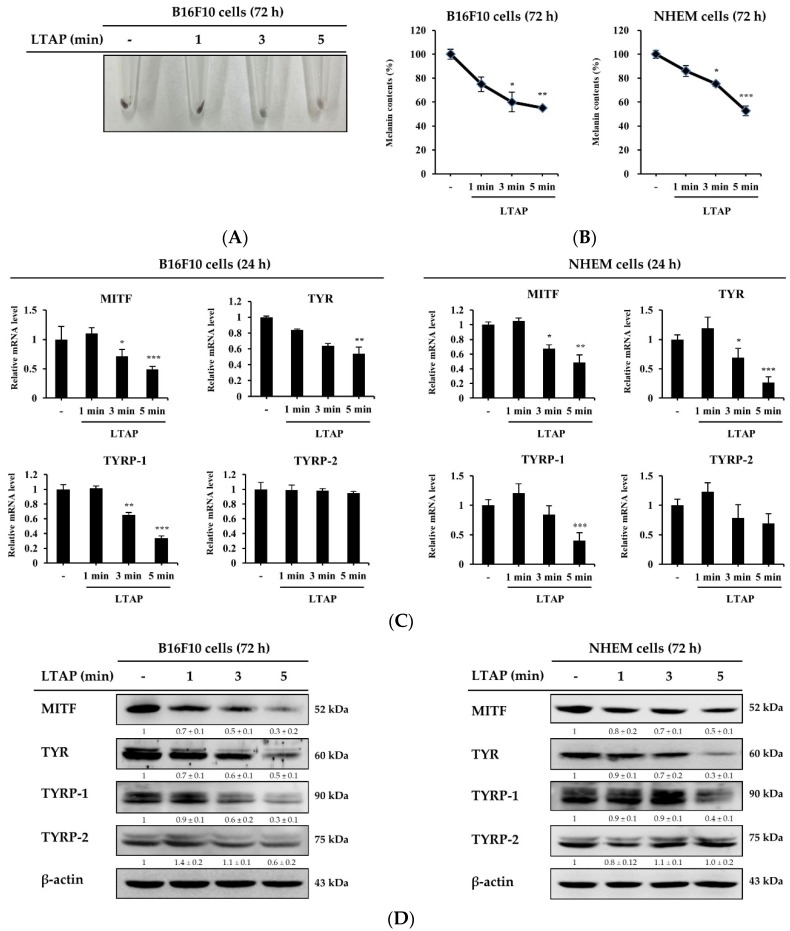
Effects of LTAP on melanogenesis in murine B16F10 cells and normal human epidermal melanocytes (NHEMs). (**A**) B16F10 cells were exposed to LTAP for 1, 3, or 5 min, and incubated further for 72 h. Cells were harvested and images were taken. (**B**) The melanin content in B16F10 cells and NHEMs was measured at 450 nm. * *p* < 0.05, ** *p* < 0.01, *** *p* < 0.001 (control vs. LTAP-exposed cells). (**C**) B16F10 cells and NHEMs were exposed to LTAP for 1, 3, or 5 min, and further incubated for 24 h. The expression levels of *microphthalmia-associated transcription factor (MITF), tyrosinase* (*TYR*) and *tyrosinase-related protein-1**, 2 (TYRP-1, 2*) were measured by qRT-PCR analysis. Data are the means ± SD and represent three independent experiments. * *p* < 0.05, ** *p* < 0.01, *** *p* < 0.001 (control vs. LTAP-exposed cells). (**D**) Cells were exposed to LTAP for 1, 3, or 5 min, and incubated for 72 h. The expression of MITF, TYR and TYRP-1, 2 was evaluated by Western blotting. β-actin was used as a loading control. Numerical values on the blots represent DU. The control was normalized to 1 DU. All data represent three independent experiments.

**Figure 6 ijms-22-01895-f006:**
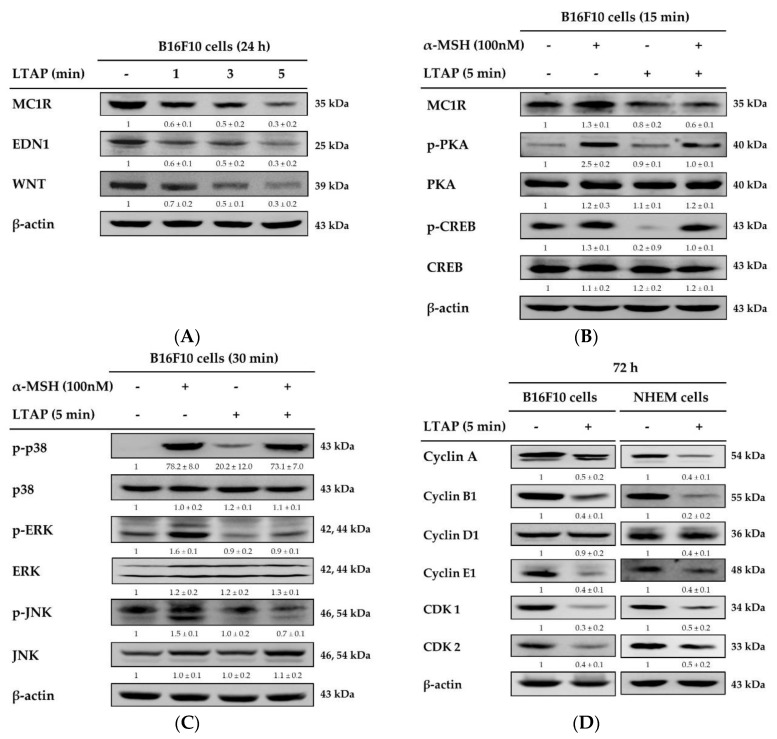
Evaluation of receptors and signals involved in melanogenesis. (**A**) B16F10 cells were exposed to LTAP for 1, 3, or 5 min, and incubated further for 24 h. The protein levels of endothelin 1 (EDN1), melanocortin 1 receptor (MC1R), and Wingless-type protein (WNT) were detected by Western blotting. β-actin was used as a loading control. (**B**,**C**) Cells were exposed to LTAP for 5 min and incubated for 3 h. Then 100 nM alpha-melanocyte stimulating hormone (α-MSH) was added and incubated for further 24 h and 15 min. Protein expression levels of MC1R were determined after 24 h of α-MSH treatment. Phosphorylation of protein kinase A (PKA), cAMP response element-binding protein (CREB) was evaluated after 15 min of α-MSH treatment. Phosphorylation of p38, extracellular signal-regulated kinase (ERK), and c-Jun-N-terminal kinase (JNK) was evaluated by Western blotting after 30 min of α-MSH treatment. (**D**) B16F10 cells and NHEMs were harvested after 72 h and the expression of Cyclin A, B1, D1, E1, and Cyclin-dependent kinase (CDK) 1 and CDK2 was evaluated by Western blotting. Numerical values on the blots represent DU. The control was normalized to 1 DU. All data represent three independent experiments.

**Figure 7 ijms-22-01895-f007:**
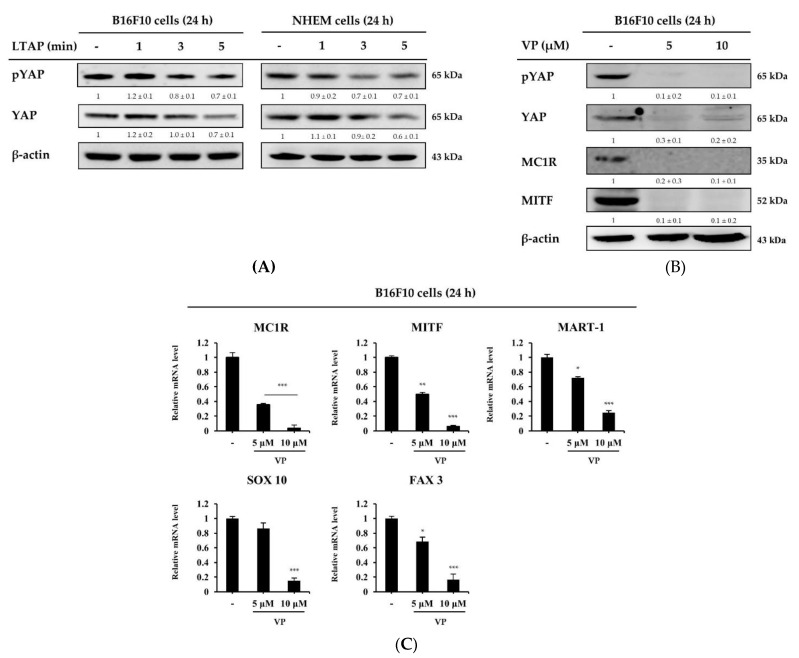
The YAP inhibitor VP decreases melanogenesis-related gene expression. (**A**) B16F10 cells and NHEMs were exposed to LTAP for 1, 3, or 5 min, and incubated further for 24 h. Cells were harvested and the levels of pYAP and YAP were evaluated by Western blotting. β-actin was used as a loading control. (**B,C**) B16F10 cells were treated with VP (5 or 10 μM), and further incubated for 24 h. Cells were harvested and the expression of pYAP, YAP, MC1R, and MITF was evaluated by Western blotting. Expression levels of *MC1R, MITF, protein melan-A (MART-1), SRY (sex determining region Y)-box 10*
*(SOX10)*, and *paired box gene 3* (*PAX3*) were measured by qRT-PCR. Data represent the mean ± SD. * *p* < 0.05, ** *p* < 0.01, *** *p* < 0.001. Numerical values on the blots represent DU. The control was normalized to 1 DU. All data represent three independent experiments.

**Figure 8 ijms-22-01895-f008:**
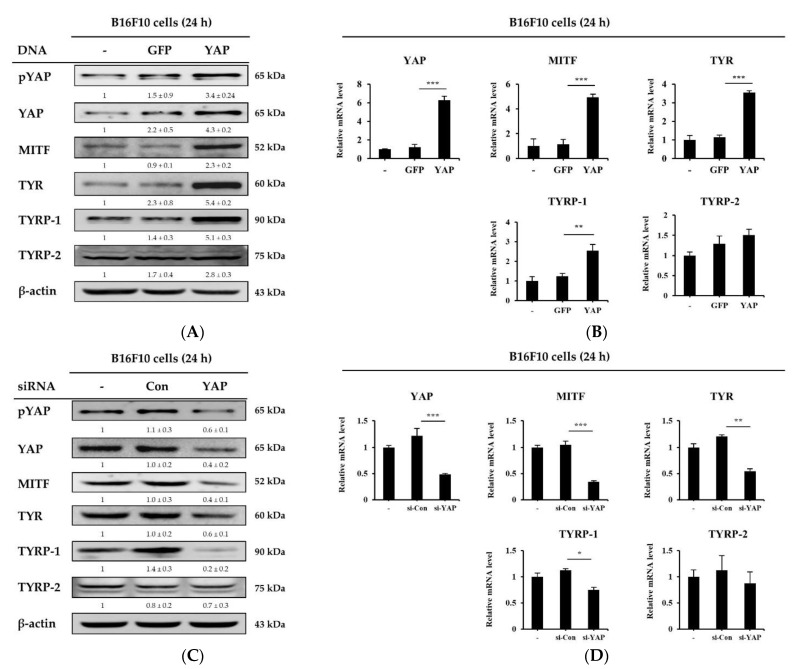
Melanogenesis-related molecules are mediated by YAP. (**A**,**B**) B16F10 cells were transfected with a plasmid overexpressing either GFP (a control) or YAP, and incubated for a further 24 h. (**A**) pYAP, YAP, MITF, TYR, and TYRP-1, 2 levels were evaluated by Western blotting. β-actin was used as a loading control. (**B**) *YAP*, *MITF*, *TYR* and *TYRP-1, 2* expression levels were determined by qRT-PCR. (**C**,**D**) B16F10 cells were transfected with either control siRNA or YAP siRNA, and incubated for 24 h. (**C**) pYAP, YAP, MITF, TYR, TYRP-1, 2 and β-actin were determined by Western blotting, and (**D**) *YAP*, *MITF*, *TYR* and *TYRP-1, 2* expression levels were analyzed by qRT-PCR. Data represent three independent experiments (means ± SD). * *p* < 0.05, ** *p* < 0.01, *** *p* < 0.001. Numerical values on the blots represent DU. The control was normalized to 1 DU.

**Figure 9 ijms-22-01895-f009:**
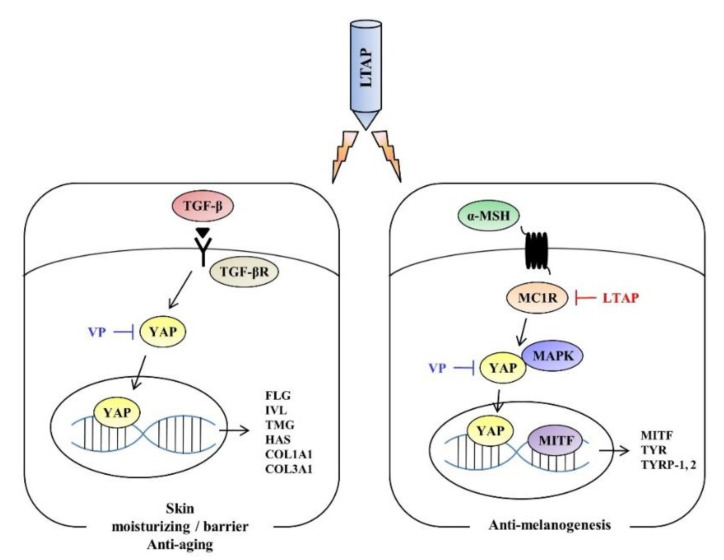
LTAP mechanism for skin moisturizing and melanogenesis. LTAP significantly increased the barrier, moisture content-related and aging factors in skin cells, and suppressed melanogenesis in melanoma cell lines through YAP regulation.

## Data Availability

The data that support the findings of this study are available from the corresponding author upon reasonable request.

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
