# Peer review of "Low-Temperature Argon Plasma Regulates Skin Moisturizing and Melanogenesis-Regulating Markers through Yes-Associated Protein"

_ijms, 2021, doi:10.3390/ijms22041895_

Round 1

Reviewer 1 Report

Kim et al. showed low-temperature argon plasma (LTAP) may regulate skin moisturizing and melanogenesis via YAP.

  • First of all, the authors did not examine the effect of LTAP in skin moisturization and only examined the effect of LTAP in regulating several molecular markers (FLG, HAS, ...) involved in skin moisturization/hydration. Please revise the title and abstract accordingly.
  • Page 1 line 17-18: In general, natural moisturizing factors include filaggrin degradation products, not transglutaminase, involucrin, and HAS. Please revise the statement.
  • I wonder why the authors did not examine the effect of YAP inhibition (verteporfin treatment) in LTAP-treated cells. By extensive series of experiments in Fig 3 (C,D), Fig 4 (B, D), Fig 7C, the authors showed the role of YAP pathway (rather than the role of LTAP) in several molecular markers.
  • Page 2 line 75: How did the authors determine the duration of LTAP exposure? (1,3 or 5 min) How was the viability of cells upon LTAP exposure? Could you provide the physiologic relevance of this exposure level in human skin in vivo?
  • I wonder if LTAP induced keratinocyte differentiation (rather than skin barrier) because the expression of FLG and IVL increased following LTAP exposure.
  • There are several typos (involucrine (-> involucrin), M (-> uM) (Fig 3), RAN (->RNA) and others) across the manuscript. Please check carefully.
  • Page 3 line 100: I wonder why the authors examined VEGF and MMP-3 in fibroblasts. MMP-1 is a more commonly examined marker in this context.
  • Page 9 line 226: That is not 'interaction' between LTAP and YAP. Please revise.

Reviewer 2 Report

Structure

The influence of LTAP on skin barrier function, skin aging and melanogenesis, via modulation of YAP expression is addressed in this paper.  These aspects of skin biology seem to be quite different, so at times the article seems disjointed and without a clear focus. Perhaps the introduction could be modified to reflect the use of LTAP as an anti-aging, skin colour therapy approach as a broader area, and then identify the specific aspects of this; to make the article more cohesive.

Introduction

In the first paragraph the referencing could be improved; there are nine lines of text before references are given in on instance and it is hard to know which reference given refers to which sentence/s. 

Please also see comments above

Results

There are many western blots and the densitometry is included as supplementary information. I understand integrating the densitometry with the western blots will increase the complexity/size of the figures; but it is also a little confusing to have to flick back and forth between two sets of figures? Perhaps it would be better to have the densitometry data in the main figure, and supporting mRNA data in supplementary?

In general there is little effort to investigate the potential functional effects of LTAP treatment and upregulation/downregulation of YAP. I think some functional assays are required e.g. data suggests that YAP affects expression of cell proliferation genes in melanocytes, thus, functional assays could also be performed to see if LTAP exposure actually led to changes in proliferation rates, or cell cycle analysis by flow cytometry for example. 

Discussion

The first paragraph of the discussion could be omitted or moved to the introduction.

Figures 3,4, 5 should be specifically referred to in the discussion. Figure 9 should also be referred to; as it presents the authors hypothesis but is then ignored in the text?

I also think suggesting LTAP as a method of reducing melanoma skin cancer is a broad reach and not supported by the data.

Methods

qPCR methods need to be expanded to include method of comparative expression (delta delta CT, I presume) and the reference gene (GAPDH). One reference gene is not considered sufficient for qPCR (MIQE guidelines) so can the authors comment on the stability of GAPDH as a reference gene and/or discuss if any other genes were considered as references?

The authors say experiments were repeated three times but not how many replicates where in each experiment? For Western blots I presume it is one, Can the authors indicate how many replicates per experiment were used? Leading on from this, ANOVA was used to determine statistical significance. If replicates were n=3 (3 separate experiments) ANOVA is not appropriate as three data points cannot have a normal distribution (an assumption required for performing an ANOVA). Non parametric statistics should be used instead.

Reviewer 3 Report

The authors studied the role of low-temperature argon plasma (LTAP) in skin moisture and melanogenesis and found that LTAP induced genes involved in skin barrier and skin moisture in HaCaT cells and normal human keratinocytes and suppressed melanogenesis in B16F10 mouse melanoma cells and normal human melanocytes. Mechanistically, LTAP seems to activate the skin barrier and moisturizing genes through YAP and inhibit melanogenesis by decreasing YAP protein. 

Major comments: 

1. The authors need to explain more about LTAP in the Introduction section. 

2. The authors need to write more details of "LTAP exposure" in the Materials and Methods section 4.2. 

Round 2

Reviewer 1 Report

My comments are well addressed. 

Author Response

Thank you very much for helping review the manuscript.